# Deep Camera–Radar Fusion with an Attention Framework for Autonomous Vehicle Vision in Foggy Weather Conditions

**DOI:** 10.3390/s23146255

**Published:** 2023-07-09

**Authors:** Isaac Ogunrinde, Shonda Bernadin

**Affiliations:** Department of Electrical and Computer Engineering, FAMU-FSU College of Engineering, Tallahassee, FL 32310, USA; bernadin@eng.famu.fsu.edu

**Keywords:** sensor fusion, object detection, deep learning, autonomous vehicles, camera–radar, adverse weather, fog, attention module

## Abstract

AVs are affected by reduced maneuverability and performance due to the degradation of sensor performances in fog. Such degradation can cause significant object detection errors in AVs’ safety-critical conditions. For instance, YOLOv5 performs well under favorable weather but is affected by mis-detections and false positives due to atmospheric scattering caused by fog particles. The existing deep object detection techniques often exhibit a high degree of accuracy. Their drawback is being sluggish in object detection in fog. Object detection methods with a fast detection speed have been obtained using deep learning at the expense of accuracy. The problem of the lack of balance between detection speed and accuracy in fog persists. This paper presents an improved YOLOv5-based multi-sensor fusion network that combines radar object detection with a camera image bounding box. We transformed radar detection by mapping the radar detections into a two-dimensional image coordinate and projected the resultant radar image onto the camera image. Using the attention mechanism, we emphasized and improved the important feature representation used for object detection while reducing high-level feature information loss. We trained and tested our multi-sensor fusion network on clear and multi-fog weather datasets obtained from the CARLA simulator. Our results show that the proposed method significantly enhances the detection of small and distant objects. Our small CR-YOLOnet model best strikes a balance between accuracy and speed, with an accuracy of 0.849 at 69 fps.

## 1. Introduction

AVs encounter several difficulties under adverse weather conditions, such as snow, fog, haze, shadow, and rain [1,2,3,4,5,6,7,8]. AVs may be affected by poor decision making and control if their perception systems are degraded by adverse weather. When water vapor condenses in the sky, it obscures the view of the surrounding area, resulting in fog. Fog can make driving unsafe because it obscures visibility. The signal-to-noise ratio (SNR) is reduced, while measurement noise rises dramatically under foggy conditions. Unsafe behavior and road accidents might be caused by sensor data that are too noisy.

Machine vision in fog can fall as low as 1000 m in moderate fog and as low as 50 m in heavy fog [9,10]. Camera sensors are one of the significant sensors used for object detection because of their low cost and the large number of features they provide [11]. In fog, the camera’s performance is limited due to visibility degradation. The quality of the image taken by a camera system can be substantially distorted by fog. In fog, lidar undergoes reflectivity degradation and a reduction in the distance measured. However, radars tend to perform better than cameras and lidars in adverse weather, since radars are unaffected by changes in environmental conditions [11,12]. Radars employ the Doppler effect to determine the distance and velocity of objects by monitoring the reflection of radio waves. With respect to object classification, radars fall short. Because radars can only detect objects, they cannot classify what kind of object they are detecting, since radar detections are far too sparse [13,14]. The sparse nature of radar point clouds collected with many vehicular radars (usually 64 points or less) might explain this [15].

However, a significant amount of research on imaging radar has been conducted over the past several years, including [16,17,18,19], which has resulted in coherent images with a centimeter-scale resolution. Detailed research on motion estimation and compensation methods for vehicle multiple-input–multiple-output synthetic aperture radar (MIMO SAR) systems was presented by Manzoni et al. [16]. The authors discuss the difficulties caused by the natural motion of the vehicle, which might result in visual abnormalities and distortions. An innovative approach to motion compensation that is based on the assessment of the platform’s motion characteristics and the subsequent compensation in the SAR processing was developed. The findings emphasize the potential of MIMO SAR for use in autonomous vehicles by demonstrating an improvement in image quality, as well as greater perception functionalities. Tebaldini et al. [17] addressed the potential of vehicle synthetic aperture radar (SAR) imaging, as well as the obstacles it faces in urban contexts. The authors investigated the distinctive features of SAR, such as its capacity to function despite unfavorable weather conditions and the fact that it can see through vegetation, which makes it appropriate for use in urban settings. Solutions to a variety of problems, such as the high computing needs for SAR processing and the demand for effective data-collecting methodologies, were provided. The findings of this study highlight the importance of performing more research and development in order to realize the full potential of SAR imaging using autonomous vehicles. Wu and Zwick [18] discussed their research on the use of synthetic aperture radar (SAR) in vehicle systems for the purpose of detecting parking lots. The authors suggested a technique for locating and categorizing parking lots that makes use of the interferometric features of SAR. The suggested method provides a high level of accuracy in the detection of parking lot boundaries because of the utilization of the coherent summation of radar echoes obtained from numerous passes. This research demonstrates the potential of SAR technology to aid autonomous vehicles in traversing complicated situations such as parking lots. Iqbal et al. [19] discussed the fundamental principles of radar imaging, including range–Doppler processing and synthetic aperture radar. In addition to this, the most important difficulties, such as reducing interference and improving resolution, were investigated. This study placed emphasis on the significance of multi-channel radar systems, sophisticated signal-processing techniques, and efficient data fusion algorithms for the purpose of the successful usage of imaging radar in self-driving automobiles. The utilization of imaging radar technology in autonomous vehicles has significant potential to enhance their perception and decision-making capabilities. Notwithstanding the advantages of imaging radar in self-driving vehicles, there are several obstacles that need to be overcome, including computational demands, data collection techniques, and interference suppression, in order to fully capitalize on its potential. The advancement of the state of the art and realization of the vision of safe and efficient self-driving cars are contingent upon further research and development in this area.

AVs are often outfitted with numerous complementary sensors to provide complementary information that helps to attain the necessary accuracy when combined. Multi-sensor fusion combines data from numerous sensors to achieve a higher object detection/classification accuracy and performance than those obtained with a single-sensor modal system [11]. Therefore, an essential subject for AVs is the combination of radar and other sensors, such as cameras. Radar–camera fusion systems can offer useful depth information for all observed objects in an autonomous driving situation. Radar sensors construct detections of nearby objects for subsequent usage, while the bounding boxes on the camera data can be used to verify and validate prior radar detections using deep-learning-based object detection methods [14].

There have been significant contributions to object detection and classification using deep learning. In addition to AV technology, object detection has found application in other fields, including surveillance and security [20], medicine [21], robotics [22,23], the military [24], etc. As outlined in [25], a deep convolutional neural network (CNN) was first utilized for image classification in 2012. However, with respect to vehicular radars, it is not uncommon for part of the observations to have incomplete, distorted, and poor-quality data. Beam obstruction, instrument malfunction, blind spots, close-to-the-ground mounting, inclement weather such as fog, and many other factors contribute to these problems. Images obtained with a camera consist of color and feature information. This feature information can be used for label classification in an object detection task. The occurrence of fog can drastically distort the feature information of an image due to atmospheric scattering and attenuation. These radar and camera problems usually lead to inaccuracies in the real-time detection of the bounding box of an object or location in an image, especially when the object is not nearby or when the object is too small under medium and heavy fog weather conditions. Thus, the application of single-sensor modal CNN-based object detection algorithms to such distorted data has proven inefficient [1,2].

YOLOv5 [26], a state-of-the-art object detection algorithm, is affected by mis-detections and false positives due to atmospheric scattering caused by fog particles. The existing deep-learning-based object detection techniques that exhibit a high degree of accuracy have a slow object detection speed in foggy weather conditions. However, several deep-learning-based object detection methods have achieved fast detection speeds at the expense of accuracy. Therefore, the problem of the lack of balance between detection speed and accuracy in foggy weather application persists. The uniqueness of radar signals and the scarcity of publicly available datasets [27] containing both camera and radar datasets [28,29,30,31,32,33] under foggy weather conditions have resulted in a limitation of AV research in this area. Very few datasets that include camera and radar information under foggy weather conditions, such as those described in [31], are available for AV research. To accommodate the needs of AVs in terms of the previously mentioned problems related to AVs’ environmental perception in fog, we make the following contributions:Using image data, we demonstrate that sensor measurements are severely impacted by atmospheric distortion in foggy conditions.We present a deep-learning-based camera–radar fusion network (CR-YOLOnet) using YOLOv5 [26] as the baseline for object detection, as shown in Figure 1. We made the following improvements to the baseline YOLOv5 to achieve CR-YOLOnet: (i) CR-YOLOnet can take its input from camera and radar sources, as compared to the single-modal system in the baseline YOLOv5. There are two CSPDarknet [34] backbone networks with which CR-YOLOnet extracts feature maps, with one each for the camera and radar channels. (ii) Using two connections inspired by the concepts of residual networks, the feature information from the backbone network is sent to the feature fusion layers. The two connections’ purpose is to improve the backpropagation of gradient in our network while minimizing feature information loss for relatively small objects in fog. (iii) We enhanced CR-YOLOnet with an attention framework to detect multi-scale object sizes in foggy weather conditions. To place more emphasis on and improve the feature representation of the features, which helps with object detection, attention modules were incorporated into the fusion layers. The attention module also helps to address the issue of high-level feature information loss so as to boost the detector’s performance.We simulated an autonomous driving environment using a CARLA [35] simulator, from which we collected both camera and radar data. We made use of both clear and foggy weather conditions for CR-YOLOnet’s training and test evaluations. To further evaluate CR-YOLOnet, we compared the matching size of the baseline YOLOv5 to that of the small, medium, and large models of our CR-YOLOnet.

This paper’s remaining sections are structured as follows: we discuss related works in Section 2, we present our methodology in Section 3, we present and discuss our results in Section 4, and Section 5 consists of the conclusion.

## 2. Related Works

### 2.1. Object Detection via Camera Only

Object detection can help to identify and determine each object instance’s spatial size and position in an image if the instances of previously defined object categories exist in the image [36]. Usually, object detection algorithms generate many potential region proposals, from which the most feasible candidates are selected [37]. The two categories of CNN-based object detection techniques are [11] (i) two-stage object detectors and (ii) one-stage object detectors. Girshick et al., in [38], proposed the first CNN-based object detection algorithm. R-CNN [38], Fast R-CNN [39], and Faster R-CNN [40] are examples of two-stage object detection algorithms. The two-stage detectors isolate the task of localizing objects using regions of interest from the task of classifying objects.

Redmon et al. [41,42,43] proposed YOLO, a one-stage detector. YOLO and its derivatives can instantly predict bounding boxes and the object class after extracting features from an input image. One-stage object detectors generate candidate regions, which are instantly used to classify and predict the target’s spatial location [1]. Backbone networks such as Feature Pyramid Networks (FPNs) [44], together with one-stage detectors such as YOLO [41], YOLO9000 [42], YOLOv3 [43], or SSD [45], have been used to detect objects via numerous detection branches in one operation, instead of predicting the potential locations and classifying them later. Because one-stage detectors do not depend on RPN for predicting potential regions, they are more efficient than two-stage detectors and are widely used for real-time object detection applications [37].

Several methods have been proposed in the literature to address autonomous driving in adverse weather conditions using cameras. Walambe et al. [46] proposed an ensemble-based method to enhance AVs’ ability to detect objects such as vehicles and pedestrians in challenging settings, such as inclement weather. Multiple deep learning models were ensembled with alternative voting techniques for object detection while using data augmentation to improve the models’ performance. In [47], Gruber et al. suggested that backscatter may be significantly reduced with gated imaging, making it a viable solution for Avs operating in severe weather conditions. In addition to offering intensity images, gated images may produce properly aligned depth data. However, eye safety standards prevent the illumination from progressing beyond sunshine, making it impossible for gated imaging to work well on extremely bright days.

Tumas et al. [48] introduced 16-bit depth modifications into YOLOv3 algorithms for pedestrian detection in severe weather conditions. While the authors employed an onboard precipitation sensor to adjust image intensity, they could not implement a real-time image enhancement for annotations collected in rain or fog. In [49], Sommer et al. used the RefineDet detection framework, which consists of some Faster R-CNN and SSD detection frameworks, for vehicle detection using traffic surveillance data. To achieve a robust detection capability, the authors proposed an ensemble network that combines two detectors, namely, SENets and ResNet-50, as the base network. However, the authors only focused on night-time and rainy scenarios. Sindagi et al. [50] proposed an unsupervised prior-based domain adaptive object detection framework for hazy and rainy conditions based on the Faster-RCNN framework. The authors trained an adaptation process using a prior-adversarial loss network to generate weather-invariant features by diminishing the weather-related data in the features. However, some improvement is required for the prior-adversarial loss network. In 2020, Hamzeh et al. [4] developed a quantitative measure of the effect of raindrop distortion on the performance of deep-learning-based object detection algorithms (including Faster R-CNN, SDD, and YOLOv3) based on a comparison between raindrop-distorted (wet) images and clear images. With the proposed quantitative measure, the amount of degradation that occurs in the detection performance of an object detector can be predicted. Liu et al. [2] conducted a study that analyzed how perception in foggy conditions impacts detection recall using a single-modal approach based on camera images. The collected camera images were characterized by deploying a Faster RCNN approach for object detection. The experimental results in [2] show that the detection recall is less than 57.75% in heavy fog conditions. This implies that a single-modal system, such as a camera-only architecture, is insufficient to handle target detection issues under adverse weather conditions.

Bochkovskiy et al. [34] proposed YOLOv4 with a CSPDarknet53 backbone and CIoU loss for evaluating prediction boxes. Jocher et al. [26] proposed YOLOv5, which uses the CSPDarknet53 backbone, the architecture of the Feature Pyramid Network (FPN) [44], and the pixel aggregation network (PAN) [51] as its neck. YOLOv5, with a large model size, tends to have a higher accuracy but low detection speed. The performance of YOLOv5 with a small model size is similar to that of YOLOv4 in terms of accuracy but faster than YOLOv4 in the speed of detection. As a result, the YOLOv5 network will serve as the baseline of our research and improvement in this study.

### 2.2. Object Detection via Fused Camera and Radar Sensors

Recently, radar signals and camera data have been combined using neural networks to accomplish various AV tasks. Radar signal representation methods include radar occupancy grid maps, radar signal projection, radar point clouds, micro-Doppler signatures, range–Doppler–azimuth tensors, etc. [11]. The numerous radar-signal-processing approaches in the literature include occupancy grid maps [52], range–velocity–azimuth maps [53], and radar point clouds [54]. As a result, several researchers [11] have suggested numerous alternative ways to represent radar signals in deep learning.

Our focus in this work is the radar signal projection method. The transformation of radar signals, such as point clouds or detections, into a two-dimensional image coordinate or a three-dimensional bird’s eye perspective is a technique known as radar signal projection. The radar, camera, and target coordinates contribute significantly to this scenario. The intrinsic and extrinsic camera calibrating matrices are used to execute the radar point cloud transformation. The resulting radar images are overlayed on the image grid. The radar image includes the radar detections and their properties, which can be fed into the DCNN network. In the literature, multiple deep-learning-based fusion methods based on vision and radar signals have been proposed.

Nabati et al. [55] suggested a technique based on a radar region proposal for object detection. Using the method reported in [55], the two-stage object detectors were eliminated, which imposed a heavy strain on region proposal creation. Radar detections were mapped onto an image plane, and the resulting image contained object proposals and anchor boxes. This approach uses radar detection instead of vision to acquire region suggestions, which saves time and effort while providing better detection results. Radar and vision sensors were combined by Chadwick et al. [56] to detect objects in the distance more precisely. It was first necessary to generate two additional imaging streams based on range and radial velocity to provide a format of the radar images on an image plane. A concatenation approach combined the radar and vision feature representations obtained from an SSD model.

Nobis et al. [57] introduced a neural-network-based object detection approach by projecting sparse radar signals onto an image vertical plane. The network was able to automatically determine the optimal level of sensor data required to increase the detection accuracy. Black-in, a novel training method that prioritizes the use of a certain sensor in a specific period to obtain better outcomes, was also introduced. Meyer et al. [58] used DCNNs to perform a low-level combination of radar point clouds and camera images to detect 3D objects. The DCNNs learn to recognize vehicles using camera images and bird’s-eye-view images generated from radar point clouds and surpass lidar camera-based systems when tested. Zhang et al. [59] proposed a radar and vision fusion system to detect and identify objects in real time. First, the object’s position and velocity were detected and obtained using radar. Subsequently, the radar data were then projected onto the corresponding image plane. A deep learning system then used ROI for target detection and identification.

John et al. [60] used the YOLO object detector to combine separate data acquired from radar and monocular camera sensors so as to better identify obstacles in inclement weather. Using two input channels, feature vectors were extracted, including one for the camera and the other for the radar. Two output channels were used to categorize the targets into groups of smaller and larger objects. A sparse radar image was created by projecting radar point clouds onto the image plane. Aiming to lessen the computational load related to real-time applications, John et al. [61] suggested a multitask learning framework based on the deep fusion of radar and camera information for joint semantic segmentation and obstacle identification.

Zhou et al. [62] proposed an object detection system based on the deep fusion of information from mm wave radar and cameras utilizing a YOLO detector. The data from the radar were utilized to generate a radar image using a single channel. The radar image was combined with the RGB camera image to produce a four-channel image that was subsequently fed into the YOLO object detector. To enhance the detection of small and minimally predictable objects, Chang et al. proposed a spatial attention module to be used in conjunction with millimeter-wave radar and vision fusion target detection algorithms [63]. None of these previous studies focused on detecting small and/or remotely distant objects under medium and heavy fog conditions.

## 3. Method

### 3.1. Sensor Calibration and Coordinate Transformation

Measurement errors grow with distance, since radar and cameras are often mounted at different locations on the ego vehicle. As a result, the shared observation region between the camera and radar requires a combined calibration effort. The vehicle’s motion defines a local right-hand coordinate system in the ego vehicle coordinate system. The local coordinate system is conveyed with the vehicle as it travels through the environment. The x-axis indicates the path of motion, whereas the y-axis is parallel to the front axle, which serves as the starting point. The camera models employ three-dimensional coordinates. For example, the camera’s x–y–z coordinate system has its origins in the camera’s viewpoint. Images acquired with camera sensors employ an image coordinate frame (x,y) and a pixel coordinate (u,v) as reference points for composition. The coordinate system of choice is the polar coordinate with respect to radar detection. The detected objects are referenced using polar coordinates. Thus, a target may be recorded as an x–y coordinate system vector in a vector space. The canonical coordinate system for radar comprises three elements: an azimuth α and the distance between the object r and the direction of the sensor’s origin. By measuring the distance of point P and its azimuth from the radar, we can estimate where P is in the world coordinate system [59,64].

The observations of the camera and radar detections can be associated using the information in a shared world coordinate system given by $\left[X;Y;Z;1\right]$. The camera calibration parameters can be used to project the radar detections onto the camera’s coordinate system and the image plane given by $\left[x;y;1\right]$. The calibration parameters of the camera can be broken down into two matrices: intrinsic and extrinsic. The intrinsic parameters’ matrix is given as [59,64]:(1)$$C=\left[\begin{array}{cccc} f_{x} & 0 & u_{0} & 0\\ 0 & f_{y} & v_{0} & 0\\ 0 & 0 & 1 & 0 \end{array}\right],$$
where $f_{x}=f/dx$ and $f_{y}=f/d_{y}$, such that f represents the focal length of the camera; $d_{x}$ and $d_{y}$ represent the physical dimensions of an individual pixel in the x–y axes’ directions, respectively; $f_{x}$ and $f_{y}$ represent the scale factors on the u and v axes; and $u_{0}$ and $v_{0}$ represent the central point offsets of the camera.

The extrinsic camera parameter can be expressed as:(2)$$\left[\begin{array}{cc} R_{3\times 3} & T_{3\times 1}\\ 0 & 1 \end{array}\right]=\left[\begin{array}{cccc} r_{11} & r_{12} & r_{13} & t_{1}\\ r_{21} & r_{21} & r_{23} & t_{2}\\ r_{31} & r_{32} & r_{33} & t_{3}\\ 0 & 0 & 0 & 1 \end{array}\right]\;,$$
where R represents the rotation parameter matrix and T is the translation parameter matrix used for mapping the radar detection point to the projection point P coordinates on the image plane. Thus, the radar detections may be mapped to their equivalent visual representations. After the mapping, the detections that fall outside the image frame are disregarded to ensure accuracy. The coordinate mapping from the world coordinate system to the image plane of the image coordinate system is as follows:(3)$$\left[\begin{array}{c} x\\ y\\ 1 \end{array}\right]=C\left[\begin{array}{cc} R_{3\times 3} & T_{3\times 1}\\ 0 & 1 \end{array}\right]\left[\begin{array}{c} X\\ Y\\ Z\\ 1 \end{array}\right],$$
where x and y represent the projection point P coordinates on the image plane.

### 3.2. Radar Detection Model

Millimeter-wave radar detects objects by sending out electromagnetic radio frequency waves in a certain direction and then analyzing the reflected signals from the environment. It is possible to determine the target’s range and velocity by monitoring the echoes’ time lag and phase change. The target’s azimuth can be obtained using directional antennas or phase comparison methods [64]. In a linear-frequency modulation continuous-wave radar signal waveform, the distance between the radar and the target causes the echo signal to have a time lag because of the propagation of electromagnetic waves. This results in a distance frequency shift $f_{d}$ for moving targets, of which the Doppler frequency shift $f_{r}$ is the outcome. Both the transmission and the echo signals will result in the generation of two differential frequencies, $f_{e}^{+}$ and $f_{e}^{-}$, at the leading and trailing edges of the frequency, respectively. The following equations can be used to determine the range R and velocity v of a target:(4)$$R=\frac{T\times c}{8B}(f_{e}^{+}+f_{e}^{-})$$
(5)$$v=\frac{c}{4f_{c}}(f_{e}^{+}+f_{e}^{-})$$
where T is the period of frequency modulation, B is the modulation bandwidth, f is the center frequency of the transmission waveform, c is the speed at which light travels, $f_{e}^{+}=f_{r}-f_{d}$, and $f_{e}^{-}=f_{r}+f_{d}$.

The phase comparison approach is utilized to provide an estimation of the azimuth. The target signal has a travel distance while it is being propagated, and as a result, the echo signal has a phase difference that corresponds to that travel distance. The target’s azimuth θ is determined using the following equation:(6)$$\theta ={sin}^{-1}(\frac{\lambda w}{2\pi d})$$
where λ is the wavelength, w is the phase shift due to the target echo signal’s propagation delay, and d is the distance between the receiving antennas.

### 3.3. Fog Imaging Model

Physical atmospheric scattering models are shown in Figure 2. The attenuation factor, transmission model, and airlight model comprise the physical atmospheric scattering model. Atmospheric scattering reduces the amount of light that may be absorbed for imaging under foggy conditions. Therefore, the target image’s object textures and edge features may be diminished. Attenuation and interference occur before the reflected light reaches the camera in foggy weather. An airtight concept allows light rays to be scattered before they reach the imaging camera. Instead of being scene lights from the item in the photograph, the transmitted lights include fog elements that obscure the images.

An image model proposed by Koschmieder [65] has frequently been used in the scientific literature [1]:(7)I(x)=J(x)t(x)+A[1−t(x)],
where I(x) denotes the picture captured by the camera, J(x) indicates the scene radiance image, t(x) denotes the transmission map, and A denotes the airlight vector, which is homogenous for each pixel in the image. The attenuation factor is represented by J(x)t(x), while the atmospheric components are represented by A[1−t(x)]. The undetermined parameters of a hazy single-input picture I are represented by the letters A, t, and J. To acquire the restored picture (recovered image) $\hat{J}$, the amount of ambient light $\hat{A}$ and transmission $\hat{t}$ can be determined using the following equation:(8)$$\hat{J}(x)=\hat{I}(x)-\frac{\hat{A}\left[1-\hat{t}\left(x\right)\right]}{\hat{t}\left(x\right)},$$

According to Narasimhan et al. [66], the visual imaging model of a foggy scenario can be regarded as the outcome of concatenating the attenuation and interference models, as shown in Figure 2. As a result of both attenuation and interference, fog can seriously degrade the quality of the image being captured in a machine. The theoretical model of the visual imaging model of a foggy scenario can also be represented as follows [2]:(9)$$E\left(d,z\right)=E_{0}\left(z\right)e^{-\beta (z)}+E_{\infty }\left(z\right){(1-e}^{-\beta (z)}),$$
where $E_{0}\left(z\right)e^{-\beta (z)}$ represents the attenuation model; $E_{\infty }\left(z\right){(1-e}^{-\beta (z)})$ represents the interference model; the light waves have a certain wavelength z; the atmospheric scattering coefficient is denoted as $E_{0}\left(z\right)$ and measures the light’s capacity to disperse per unit volume; the depth of the scene is represented as d; the scattering coefficient is denoted as β; and $\beta \left(z\right)$ indicates the intensity of the target obstacle’s light as it is scattered through the atmosphere and reaches the camera.

As mentioned earlier, the scattering impact of incoming light on airborne particles in the atmosphere will reduce the intensity of the light that ultimately reaches the camera [11]. We consider the relationship between the depth *d* of the scene and transmission *t*. We also consider the effect of image degradation due to the attenuation of the visibility of the image. Consider an observer (imaging camera) at distance *d*(*x*) from a scene point at position x. The relationship between the transmission t and depth d is expressed in the following equation [67]:(10)$$t\left(x\right)=exp\left(-\int_{0}^{d(X)}\beta \left(z\right)dz\right),$$
where $\int_{0}^{d\left(x\right)}\;,$ is the distance between the imaging camera and the scene point at x, and β represents the atmospheric scattering coefficient. If the atmosphere exhibits homogeneous physical properties, the scattering coefficient β will be the spatial constant. Therefore, Equation (4) can be rewritten as:(11)$${t\left(x\right)=e}^{-\beta d\left(x\right)},$$

The transmission ${t\left(x\right)=e}^{-\beta d\left(x\right)}$ illustrates the unscattered part of the light that reaches the camera. From Equation (11), we can express $d\left(x\right)$ as follows:(12)$$d\left(x\right)=-\frac{lnt(x)}{\beta },$$

Equation (12) implies that the depth can be calculated up to an unknown scale if the transmission can be estimated [67]. The visibility distance, measured in meters, is the maximum distance at which black and white objects lose their distinct contrast. As the distance increases in fog, a black and white object seems to become a uniform gray color to the human eye. Therefore, the standard maximum contrast ratio is 5% [68].

Figure 3a depicts clear, foggy images collected during a real-time autonomous driving simulation at 100 m and 25 m visibility distances. Figure 3b illustrates the contrast between the grayscale of the clear and foggy images at the visibility distances of 100 m and 25 m. The information regarding an image’s colors and features can be clearly revealed when the image is converted to grayscale. The information regarding the image’s features can be extracted and used for classification purposes in an object detection task. As shown in Figure 3b, the range of the grayscale of the clear-day image is from around 0 to 250. The grayscale of the foggy images at the 100 m and 25 m visibility distances is highly concentrated between 30 and 210 and between 100 and 250, respectively. As a result, the detection of objects can be negatively affected by fog, because it drastically distorts the image’s feature information [3].

Figure 3c shows a simulation of a real-time autonomous driving scene that lasted for 12 s in clear (no fog) and heavy fog conditions with a visibility distance of 25 m. Because sensor measurement noise tends to increase significantly in fog, the signal-to-noise ratio (SNR) value decreases dramatically. Figure 3c illustrates a higher SNR value in the no-fog scene and a much lower SNR value in the heavy fog scene.

### 3.4. The Baseline YOLOv5 Model

YOLO is a cutting-edge, real-time object detection algorithm, and YOLOv5 [26] is built on earlier versions of the YOLO algorithm. YOLO is one of the most effective object detection methods available, with a notable performance, yielding state-of-the-art results on datasets such as the Microsoft COCO [69] and Pascal VOC [70].

The backbone, neck, and head sections are the three fundamental components of the baseline YOLOv5 network, as shown in Figure 4. The functionality of the backbone section involves the extraction of relevant feature data from the input images. The neck combines the collected features to create three different scales of feature maps used by the head to detect objects in the image. The YOLOv5 backbone network is CSPDarknet, and the neck consists of the FPN (Feature Pyramid Network) structure and PAN (Spatial Pyramid Pooling) structure.

(i)
*Backbone:*


In YOLOv5, Darknet [43] was merged with a cross-stage partial network (CSPNet) [71], resulting in CSPDarknet. The CSPDarknet is composed of convolutional neural networks that use numerous iterations of convolution and pooling to generate feature maps of varying sizes from the input image. As a solution to the issues caused by the repetition of gradient information in large-scale backbones, CSPNet incorporates the gradient transitions into the feature map. Thus, reducing the model’s size and the number of parameters and floating-point operations per second guarantees fast and accurate inference. For an object detection task in fog, it is crucial to have a compact model size, fast detection speed, and high accuracy. The backbone generates four distinct levels of feature maps, including 152 × 152 pixels, 76 × 76 pixels, 38 × 38 pixels, and 19 × 19 pixels.

The backbone focus module (Figure 5a) is used for slicing operations. The purpose of the focus is to improve feature extraction during downsampling. Convolution, batch normalization, and the leaky ReLU activation function (AF) are all sub-modules of the CBL module. YOLOv5 implements two distinct cross-stage partial networks (CSP), as shown in Figure 5b. Each has a specific function; one is for the neck of the network, and the other is for the backbone. The CSP network uses cross-layer communication between the front and back layers to shrink the model size while preserving its accuracy and increasing inference speed. The feature map of the base layer is divided into two distinct parts: the main component and a skip connection. These two parts are then joined through transition, concatenation, and transition to reduce the amount of duplicate gradient information as effectively as possible. Regarding CSP networks, the difference between the backbone and the neck is that the latter uses CBL modules instead of residual units.

Maximum pooling with varying kernel sizes is carried out using the Spatial Pyramid Pooling, or SPP, module [72], as shown in Figure 5c. The features are fused through concatenation. The SPP module undertakes dimensionality reduction procedures to convey image features at a higher degree of abstraction. Pooling reduces the feature map’s size and the network’s computational cost while extracting the essential features.

(ii)
*Neck:*


The feature maps from each level are fused by the neck (FPN and PAN) network to learn more contextual information and lessen the amount of data lost in the process. The low-level structures present in the feature maps near the image layer render them ineffective for precise object detection. Feature Pyramid Network (FPN) was designed to extract features to maximize detection speed and accuracy. FPN enables a top-down mechanism to generate higher-resolution layers from significant, robust semantic feature layers. The PAN architecture effectively transfers localization features through a down-top mechanism from lower to higher feature maps to improve the position accuracy of objects in the image. Thus, feature maps are generated on three different scales on three feature fusion layers.

(iii)
*Detection Head:*


The detection head consists of convolution blocks that take the three different scales of the feature maps from the neck layer. Through convolution, the detection head yields three distinct sets of detections with resolution levels of 76×76×255, 38×38×255, and 19×19×255. Every grid unit in a feature map correlates to a larger portion of the original image as the feature map’s resolution decreases. This implies that the 76×76×255 and 19×19×255 feature maps can adequately detect small and large objects.

### 3.5. Attention Mechanism

Numerous studies discovered that when deep CNN reaches a particular depth, it degenerates [73]. Studies have shown that networks’ performance does not necessarily improve significantly with depth but can substantially increase computational costs throughout the training phase [74]. Therefore, the attention mechanism was created to train networks in order to prioritize and devote more attention to relevant feature information while down-ranking that which is less relevant [75]. The attention mechanism informs CNNs where to focus their attention and improves the feature representational power of the features, which helps with object detection tasks. The human eye provides proof that attention mechanisms are crucial for collecting relevant data [76]. This behavior has prompted several studies [76,77,78,79,80] aiming to improve convolutional neural networks’ efficiency in image classification problems by including an attention mechanism. In 2018, Woo et al. [78] proposed the Convolutional Block Attention Module (CBAM), which integrates spatial and channel attention into a single lightweight mechanism. A considerable performance boost can be achieved with ECA-Net [80], proposed by Wang et al. in 2020. ECA-Net is an efficient channel attention mechanism that can collect information regarding cross-channel relationships.

CBAM [78] was designed to simultaneously capture both channel and spatial attention modules. Since the channels of feature maps are treated as feature detectors, the channel attention module focuses on the most important features in the input images. This makes the channel attention module an essential application for an image-processing task such as object detection in fog. Average pooling and max-pooling were employed to aggregate the spatial information of the input feature to obtain average-pooled and max-pooled features. For an input feature map $F\in R^{\left(C\times H\times W\right)}$, individual channel weights are estimated, where the number of channels is C and the length and width of the feature map in pixels are H and W, respectively. The weighted multiplication of channels is useful for drawing more attention to the primary channel features. A shared network (multi-layer perceptron) with one hidden layer is used for both the average-pooled and max-pooled feature descriptors. The element-wise summation of the output vector of both descriptors then generates the channel attention weight map $M_{c}\in R^{\left(C\times 1\times 1\right)}$ using Equation (13). The channel-refined feature maps are obtained through the element-wise multiplication of the original feature map and $M_{c}\in R^{\left(C\times 1\times 1\right)}$:(13)$$M_{c}(F)=\;\;\sigma \left(MLP\left(AvgPool\left(F\right)\right)+MLP\left(MaxPool\left(F\right)\right)\right)\;=\sigma \left(W_{1}\left(W_{0}\left(F_{avg}^{c}\right)\right)+W_{1}\left(W_{0}\left(F_{max}^{c}\right)\right)\right),$$
where σ is the sigmoid activation function, $W_{1}$ and $W_{0}$ are the multi-layer perceptron weights, $F_{avg}^{c}$ denotes the average-pooled features, and $F_{max}^{c}$ denotes the max-pooled features.

Next, the spatial component uses the channel-refined features from the channel submodule to generate a 2D spatial attention map. The element-wise multiplication of the spatial attention weight map and the input channel attention feature map generates the final refined feature map through the attention mechanism [81]. The spatial attention module pays the most attention to the object’s position in the image frame. This is achieved by combining the spatial features in an individual space using the weighted sum of spatial features. The overall refined features are obtained by multiplying the channel-refined features from the 2D spatial attention map. For a channel-refined feature map $F_{c}\in R^{\left(C\times H\times W\right)}$, the convolution of the average pooling and max-pooling using a 7×7 filter size gives the spatial attention weight map $M_{s}\in R^{\left(1\times h\times w\right)}$, as shown in Equation (14):(14)Fs=1c∑i∈cFc(i)+Fc(i)i∈cmaxMS=σf7×7Fs,
where σ is the sigmoid activation function and $f^{7\times 7}$ is a convolution with a 7×7 filter size.

However, to lessen the number of parameters, CBAM uses dimensionality reduction to help to manage the model’s complexity. Nonlinear cross-channel relationships are captured throughout the dimensionality reduction process. The dimensionality reduction can lead to an inaccurate capture of the interaction between channels. We adopted the ECA-Net approach [80] to solve this problem. ECA-Net uses global average pooling (GAP) to aggregate convolution features without reducing dimensionality. This is accomplished by increasing the number of parameters to a very modest degree while successfully gathering details regarding cross-channel interactions and gaining a substantial performance improvement. To understand channel attention, the ECA module adaptively estimates the kernel size K. It then conducts a 1D convolution and applies a sigmoid function σ. The kernel size K can be adaptively determined as follows:(15)$$K=\psi \left(C\right)={\left|\frac{\log_{2}\left(c\right)}{\gamma }+\frac{b}{\gamma }\right|}_{odd}.$$
where ${\left|t\right|}_{odd}$ represents the nearest odd number of t, the kernel size K can be determined using mapping ψ, and the number of channels (channel dimension) is denoted as C. γ is set to 2, and b is set to 1.

In this work, we combined ECA-Net and CBAM to achieve a powerful attention mechanism, as illustrated in Figure 6. We incorporated the combined ECA-Net/CBAM attention mechanism into the fusion layers of our proposed camera–radar fusion network (CR-YOLOnet) shown in Figure 7. The attention mechanism helps to draw more attention to and improve the feature representation of the features, which helps with object detection. We enhanced CR-YOLOnet with an attention framework to detect multi-scale object sizes in foggy weather conditions. ECA-Net handled the channel submodule operations, while CBAM handled the spatial submodule operations. The ECA-Net module is effectively trained on the input feature maps following a 1D convolutional GAP which generates the updated weight.

The channel-refined feature maps are produced through the element-wise multiplication of input feature maps and the updated weight. The output of the ECA module is sent to CBAM’s spatial attention module, which generates a 2D spatial attention map. The element-wise summation of the original input feature map and 2D spatial attention map is performed to obtain a residual-like architecture. The ReLU activation function is applied to the aggregated feature map to generate the final feature map, which sent to the detection head layer shown in Figure 7.

### 3.6. Proposed Camera–Radar Fusion Network (CR-YOLOnet)

We present our proposed network, called CR-YOLOnet, in Figure 7, a deep learning multiple-sensor fusion object detector based on the baseline YOLOv5 network. To develop CR-YOLOnet, we made several adjustments to the baseline YOLOv5 model. Our CR-YOLOnet can take its input from camera and radar sources, as compared to the single-modal system in the baseline YOLOv5. There are two CSPDarknet backbone networks with which CR-YOLOnet extracts feature maps, with one each for the camera and radar sensors.

The feature information from the backbone network is sent to the feature fusion layers through two connections, illustrated as round-dotted lines. The concepts of residual networks inspired the connections to improve the backpropagation of gradient in our network, prevent gradient fading, and minimize feature information loss for relatively small objects in fog.

As previously mentioned in Section 3.4, we included the combined ECA-Net/CBAM attention mechanism in the fusion layers of CR-YOLOnet. The purpose of the attention mechanism is to enhance the capacity of CR-YOLOnet to detect multi-scale object sizes in medium and heavy fog weather conditions, especially small objects that are not nearby.

The detection head is composed of convolution blocks and utilizes all three scales of the feature maps in the neck layer. The two-dimensional convolution allows the detection head to produce three unique sets of detections, each having a resolution level of 80×80×12, 80×80×12, and 20×20×12, respectively. The depth is 12 because the number of object classes is 7, the confidence level is 1, and the positional parameters are 4 in number, the total sum of which is 12.

### 3.7. Loss Function

Three components comprise the loss function: (i) the bounding box (position) loss, (ii) confidence loss, and (iii) classification loss. The bounding box loss function can be calculated when the intersection of the prediction box and the actual box is larger than the set threshold. The confidence loss and classification loss calculations are made when the object center enters the grid.

#### 3.7.1. Bounding Box Loss Functions

We employed the complete intersection of union (CIoU) loss for bounding box regression [82]. This is because the CIoU combines the following: (i) the overlap region between the predicted bounding box and the ground truth bounding box, (ii) the central point distance between the predicted bounding box and the ground truth bounding box, and (iii) the aspect ratio of the predicted bounding box and the ground truth bounding box. The CIoU approach combines these three components to improve the accuracy of the average precision (*AP*) and average recall (AR) for object detection while achieving a faster convergence.

The CIoU loss function in Equation (16) builds on the distance intersection of union (DIoU) loss [82] by enforcing a penalty term $R_{CIoU}$ for the box aspect ratio given in Equation (17):(16)$$L_{CIoU}=1-IoU+R_{CIoU}$$
(17)$$R_{CIoU}=\frac{\rho^{2}(b,\;b^{gt})}{c^{2}}+\alpha \upsilon$$
(18)$$\alpha =\frac{\upsilon }{1-IoU+\upsilon }$$
(19)$$\upsilon =\frac{4}{\pi^{2}}{\left(arctan\frac{w^{gt}}{h^{gt}}-arctan\frac{w}{h}\right)}^{2}$$
where α is the weight function, a trade-off parameter that gives the overlap region factor a higher priority for regression, especially for non-overlapping cases; υ helps to measure the consistency or similarity of the aspect ratio between the bounding boxes; b and $b^{gt}$ are the central points of the predicted bounding box B and the ground truth bounding box $B^{gt}$; and the widths and heights of the predicted bounding and the ground truth bounding boxes are denoted as w and h and as $w^{gt}\;$and $h^{gt}$, respectively.

#### 3.7.2. Confidence Loss and Classification Loss Functions

The confidence loss function $L_{obj}$ is as follows:(20)$$L_{obj}=\sum_{i=0}^{s\times s}\sum_{j=0}^{b}I_{ij}^{obj}\left[{\hat{C}}_{i}\log\left(C_{i}\right)+\left(1-{\hat{C}}_{i}\right)(1-\log\left(C_{i})\right)\right]\;{-\lambda }_{noobj}\sum_{i=0}^{s\times s}\sum_{j=0}^{b}I_{ij}^{noobj}[{\hat{C}}_{i}\log\left(C_{i}\right)+(1-{\hat{C}}_{i})(1-\log\left(C_{i})\right)]$$

The classification loss function $L_{cls}$ is as follows:(21)$$L_{cls}=\sum_{i=0}^{s\times s}\sum_{j=0}^{b}I_{ij}^{obj}\sum_{c\in classes}\left[{\hat{P}}_{i}\left(c\right)\log p_{i}\left(c\right)+(1+{\hat{P}}_{i}\left(c\right))\log\left(1-p_{i}(c)\right)\right]$$
where $I_{ij}^{obj}$ represents the object detected by the $j^{th}$ boundary of the grid cell, s×s denotes the number of grid points, b denotes the number of anchors associated with each grid, c denotes the number of categories, p represents the probability of categories, C denotes the box confidence score in cell i, ${\hat{C}}_{i}$ denotes the box confidence score for the predicted object, and $\lambda_{noobj}$ denotes the weight representing the predicted loss of confidence in the bounding box in the absence of an object.

Therefore, the overall loss function is given as follows:(22)$$Loss=\sum_{i=0}^{s\times s}L_{CIoU}+L_{obj}+L_{cls}$$

## 4. Experimental Results

### 4.1. Dataset

In this work, we used the CARLA [35] simulator to create a simulated environment for autonomous driving, from which we collected camera and radar data. Seven (7) different types of common road participants were included in our datasets. Since the camera observations and radar detections were associated, the radar detections were sparsely overlapped as white dots on the camera image, as shown in Figure 8.

Figure 9 illustrates a sample of our CARLA dataset showing clear day conditions and varying fog levels. According to visibility, we classified foggy weather into one of four conditions, as shown in Table 1. We determined the visibility for each individual traffic scenario in the experiment based on [2,9,10]. In clear weather, the visibility distance is greater than 1000 m, while that in light fog is 500–800 m, in medium fog is 300–500 m, and in heavy fog is 50–200 m. The total number of images is 25,000, with 80% (20,000) belonging to the training set and the remaining 5000 for testing and verification. We used clear and foggy weather conditions for CR-YOLOnet’s and YOLOv5’s training and testing evaluations. Figure 10 shows the distribution of various object classes, including those of bicycle, bus, car, motorcycle, person, traffic light, and truck.

### 4.2. Experimental Platform and Training Parameters

The PyTorch version 1.9.0 was used to conduct an experiment using Pythonversion 3.9.6. The hardware and software settings were as follows: graphics card: Nvidia GeForce RTX 2070 with Max-Q Design; RAM: 16 gigabytes of memory; and CPU: Intel Core 17-8570H, 2.2 GHz, six cores. Table 2 illustrates the parameters for the three different model sizes (small, medium, and large), with which the CR-YOLOnet and baseline YOLOv5 models were trained. With around only 7.5 million parameters, YOLOv5s is a small but fast model, making it well-suited for inference on the central processing unit.

The YOLOv5m model is considered medium-sized, with its 21.5 million parameters, because it strikes an outstanding balance between speed and accuracy. Among the YOLOv5 derivatives, YOLOv5l is the largest, with a total of 46.8 million parameters. It is efficient for the detection of small objects. The CR-YOLOnet was trained on both image and radar data, using only image data for YOLOv5. To begin with, the rate of learning steadily increased, and then it gradually decreased. The network’s utilization of the pre-training rate caused the increase in the learning process in the beginning. Each model was trained using the clear only, fog only, and clear + fog datasets for 300 epochs with a batch size of 64, weight decay of 0.00025, a learning rate of 0.0001, and a learning rate momentum of 0.821 using Adam optimization.

### 4.3. Evaluation Metrics

Deep learning can be evaluated using a variety of metrics, including accuracy, the confusion matrix, precision, recall, average precision, mean average precision (map), intersection union ratio, and average precision. In this work, we use the same set of evaluation metrics as that of the COCO dataset [69], including precision, recall, and the average precision (*AP*) for small (*AP*s), medium (*AP*m), and large (*AP*l) object areas. We also estimated the *F*1 score and mean average precision (*mAP*) thresholds at 0.5. We compared the performance of our CR-YOLOnet to that of the baseline YOLOv5 in clear, light, medium, and foggy environments for small, medium, and large model sizes. Equations (23)–(27) describe the evaluation metrics.

Precision P can be expressed as:(23)$$P=\frac{TP}{TP+FP}$$

Recall R can be expressed as:(24)$$R=\frac{TP}{TP+FN}$$

The *F*1 score can be expressed as:(25)$$F1=\frac{2(P\times R)}{P+R}$$
where *TP* denotes the outcome that occurs when the category of an object is accurately identified in an image, *FP* represents the outcome that occurs when the category of an object is inaccurately identified in an image, and *FN* is the outcome that occurs when an attempt to identify an object in an image fails.

The average precision (AP) is the area under the Precision−Recall curve with values between 0 and 1, and it is expressed in Equation (26):(26)$$AP=\int_{0}^{1}P(R)dR$$

The mean average precision (*mAP*) is the mean of all N categories evaluated in the dataset, and it can be estimated as follows in Equation (27):(27)$$mAP=\frac{1}{N}\sum_{i=1}^{N}AP_{i}$$

### 4.4. Training Results and Discussion

To ensure the algorithm’s detection efficiency, our improved method (CR-YOLOnet) is compared to the baseline YOLOv5. A contrast of the changes in *mAP* that occur throughout the training process for our CR-YOLOnet and the YOLOv5 (small, medium, large) models can be seen in Figure 11. Each CR-YOLOnet model was trained on the radar and image (clear only, fog only and fog + clear) training sets. When compared to the YOLOv5 network, the rise in *mAP* exhibited by the CR-YOLO network was stable and much quicker due to its multi-sensor integration advantage.

The large CR-YOLOnet model, as shown in Table 3, clearly achieves the highest performance, with an F1 of 0.861, recall of 0.885, precision of 0.914, and *mAP* of 0.896. However, the network that strikes the best balance between accuracy and speed is our small CR-YOLOnet model, with an *mAP* of 0.849 and 69 frames per second.

### 4.5. Testing Results and Discussion

Observing the model’s performance in various clear day and foggy weather conditions is essential for establishing its reliability. Table 3, Table 4 and Table 5 show the comparison of detection *AP* for small, medium, and large object areas and *mAP* at IoU=0.5. The comparison was made between the large (Table 4), medium (Table 4), and small (Table 5) model sizes under clear, light, medium, and heavy fog conditions.

In Table 4, CR-YOLOnet trained on the clear day datasets performed the best under clear weather conditions, with an APs of 0.928 and APl of 0.989. However, CR-YOLOnet trained on the clear + fog datasets performed better than the other five models, with the highest *mAP* of 0.892 for clear and foggy conditions. An improvement of 11.78% in *mAP* was observed when compared to YOLOv5 trained on the clear + fog datasets, with an *mAP* of 0.798. Table 4 shows that CR-YOLOnet, with an *AP*s of 0.912 and *AP*l of 0.975, performed the best in clear weather when it was trained on the clear day datasets. Out of the six models tested, CR-YOLOnet trained on the clear + fog datasets had the most significant (*mAP*) of 0.867. An improvement of 13.33% in *mAP* was noted when compared to YOLOv5 trained on the clear + fog datasets, with an *mAP* of 0.765.

In Table 6, CR-YOLOnet trained on the clear + fog datasets outperformed the other five models in almost all the metrics. In Table 7, we illustrate the comparison of the detection *AP* per object class. The CR-YOLOnet trained on the clear + fog datasets outperformed the other five models for each object class. However, compared to the large (Table 4) and medium (Table 5) models, the CR-YOLOnet trained on the clear + fog datasets in Table 6 struck a balance between accuracy and speed, with an *mAP* of 0.847 and speed of 72 FPS for both clear and foggy circumstances. This implies that our small CR-YOLOnet model trained on the clear + fog datasets has the best capacity to accurately detect small objects in fog without a trade-off of speed.

Thus, in Figure 12, we compare the qualitative results of our small CR-YOLOnet and the medium YOLOv5 model, with both models trained on the clear + fog datasets. We selected the medium YOLOv5 model trained on the clear + fog datasets because it struck a balance between speed and accuracy, as illustrated in Table 5.

Figure 12a shows the input data with varying visibility and proximity of the close objects at approximately 50 m and most distant objects at 300 m. Figure 12b,c shows the detection results of the medium YOLOv5 and small CR-YOLOnet models, respectively. Both models could detect objects in close proximity. However, only our small CR-YOLOnet model trained on the clear + fog datasets could detect objects beyond 100 m in medium fog and beyond 75 m in heavy fog conditions.

## 5. Conclusions

In this paper, we introduced an enhanced YOLOv5-based multi-sensor fusion network (CR-YOLOnet) that fused radar object identification with a camera image bounding box to locate and identify small and distant objects in fog. We transformed the radar detections by mapping them onto two-dimensional image coordinates and projected the resulting radar image onto the camera image. Using image data, we demonstrated that atmospheric distortion has a negative impact on sensor data in fog. We showed that our CR-YOLOnet, in contrast to the single-modal system used in the baseline YOLOv5, is capable of receiving data from both camera and radar sources. CR-YOLOnet utilizes two different CSPDarknet backbone networks for feature map extraction, one for the camera sensors and the other for the radar sensors.

We emphasized and improved critical feature representation required for object detection using attention mechanisms and introduced two residual-like connections to reduce high-level feature information loss. We simulated autonomous driving instances under clear and foggy weather conditions using the CARLA simulator to obtain clear and multi-fog weather datasets. We implemented our CR-YOLOnet and the baseline YOLOv5 in model configurations of three sizes (small, medium, and large). We found that both the small CR-YOLOnet and medium YOLOv5 trained on clear + fog datasets struck a balance between speed and accuracy, with an *mAP* of 0.847 and a speed of 72 FPS. There was an improvement of 24.19% in *mAP* when compared to YOLOv5 trained on the clear + fog datasets, with an *mAP* of 0.765. However, the performance of CR-YOLOnet was significantly improved, especially in medium and heavy fog conditions. Since the large YOLOv5 model is more efficient for the detection of small objects, in the future, we could optimize the speed of our large CR-YOLOnet by reducing the dimensions of the input data using half-precision floating points, which lower the memory usage in neural networks, and enhance the backbone network with an attention mechanism, etc., without a trade-off of accuracy.

## Figures and Tables

**Figure 1 sensors-23-06255-f001:**
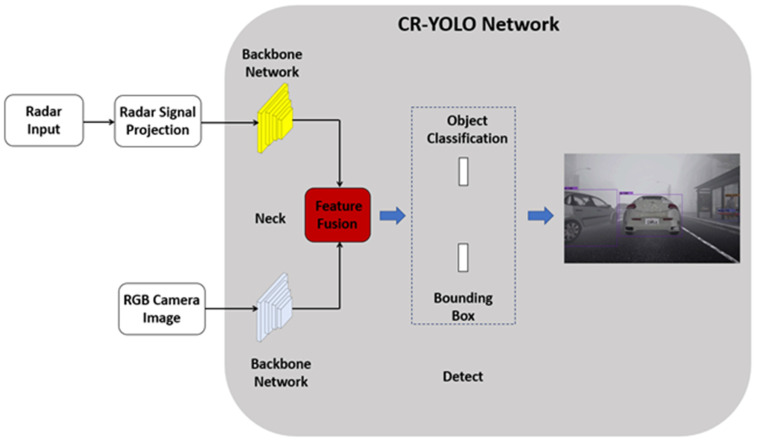
The proposed camera–radar fusion network (CR-YOLOnet).

**Figure 2 sensors-23-06255-f002:**
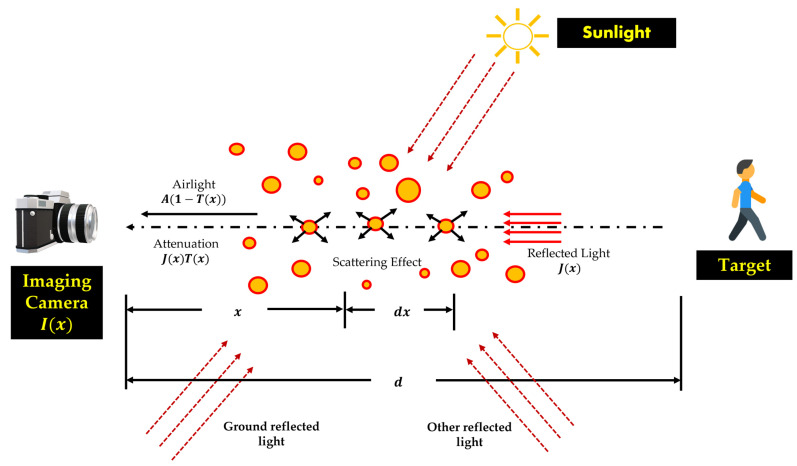
An atmospheric scattering phenomenon of a foggy imaging model.

**Figure 3 sensors-23-06255-f003:**
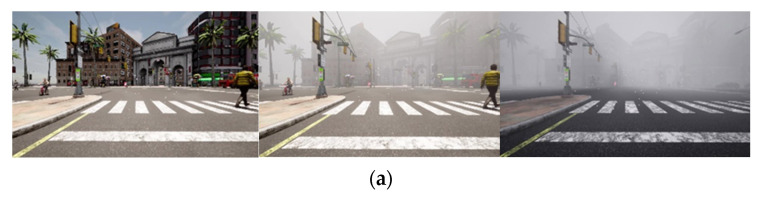
(**a**) Clear and foggy images at 100 m and 25 m visibility distances. (**b**) The comparison of grayscales for clear and foggy images at 100 m and 25 m visibility distances. (**c**) The comparison of the SNR values for clear (no fog) and heavy fog conditions at a visibility distance of 25 m.

**Figure 4 sensors-23-06255-f004:**
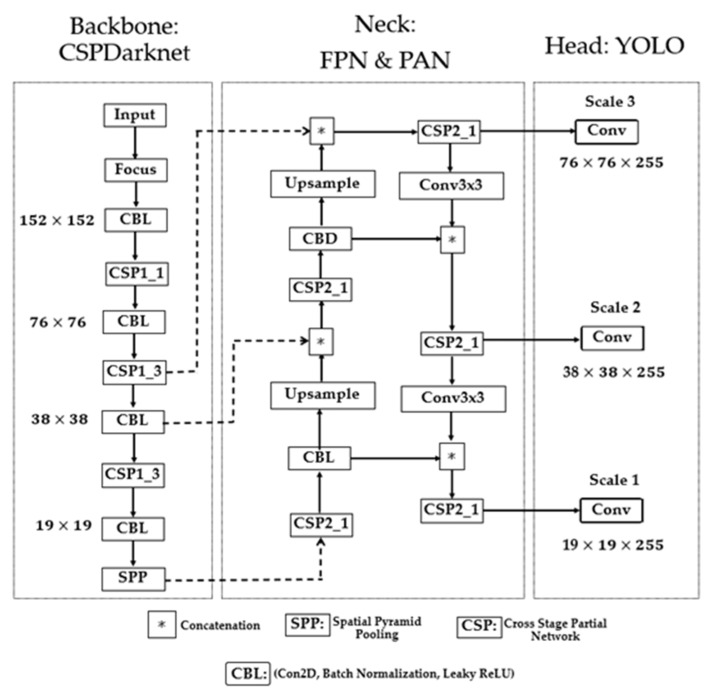
The baseline YOLOv5 architecture.

**Figure 5 sensors-23-06255-f005:**
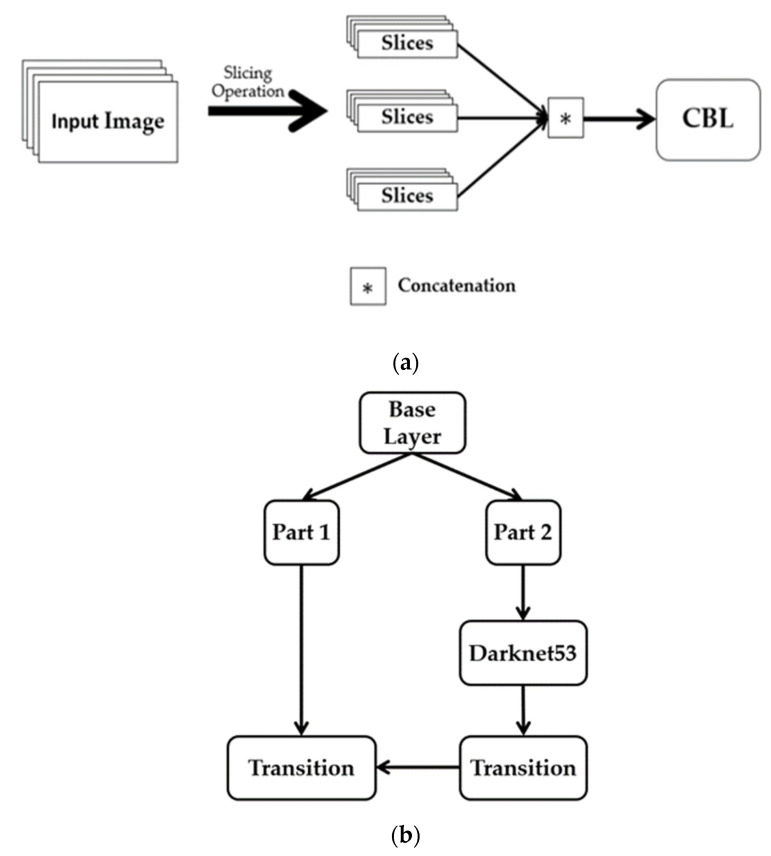
Illustration of (**a**) the focus architecture, (**b**) the CSPDarkNet53 architecture, and (**c**) the SPP architecture.

**Figure 6 sensors-23-06255-f006:**
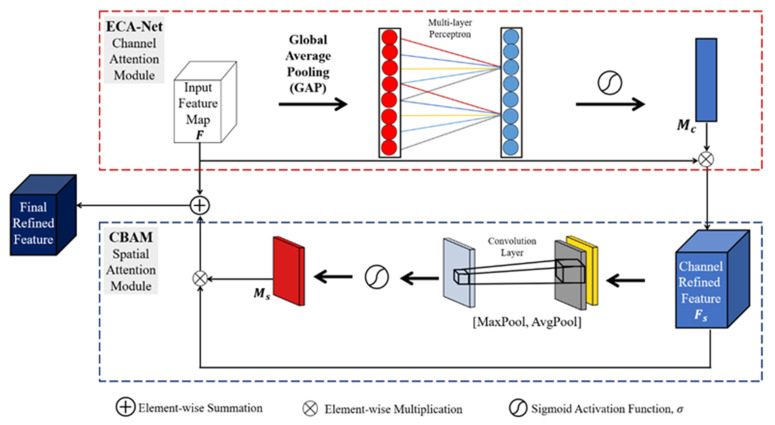
The attention module architecture: the combination of the ECA-Net and CBAM attention submodules to develop a complete channel and spatial attention mechanism.

**Figure 7 sensors-23-06255-f007:**
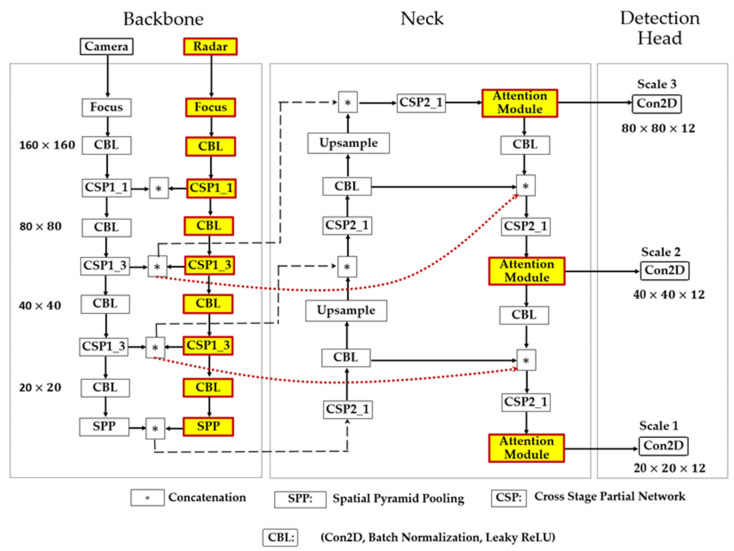
The architecture of our proposed CR-YOLOnet with the attention module incorporated.

**Figure 8 sensors-23-06255-f008:**
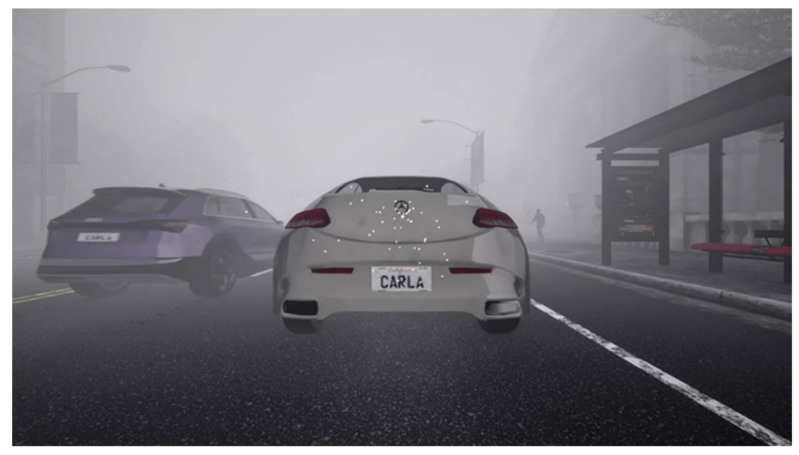
Camera and radar data obtained from the CARLA simulator with radar data overlayed as white dots.

**Figure 9 sensors-23-06255-f009:**
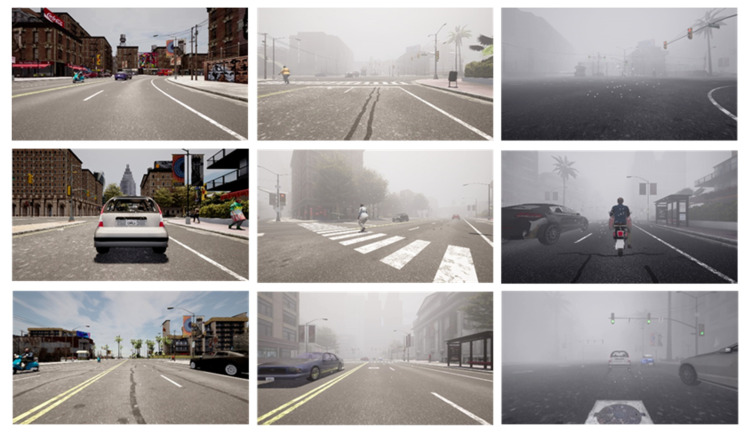
A sample of our CARLA dataset showing a clear day in the far-left column and varying levels of fog in both columns to the right.

**Figure 10 sensors-23-06255-f010:**
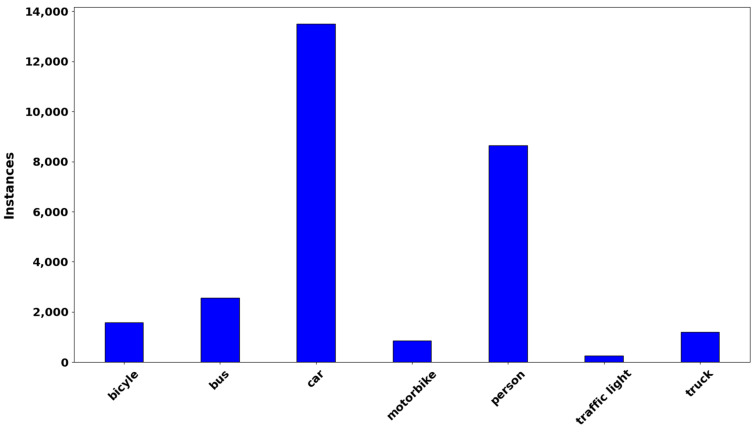
The distribution of various objects classes.

**Figure 11 sensors-23-06255-f011:**
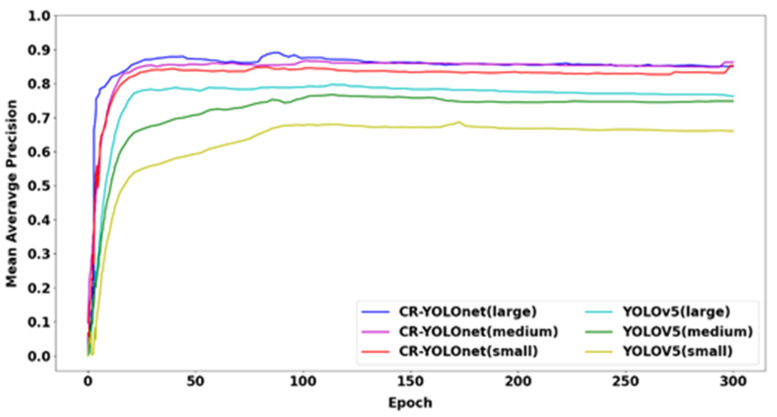
Comparison of *mAP* (0.5) between CR-YOLOnet and YOLOv5.

**Figure 12 sensors-23-06255-f012:**
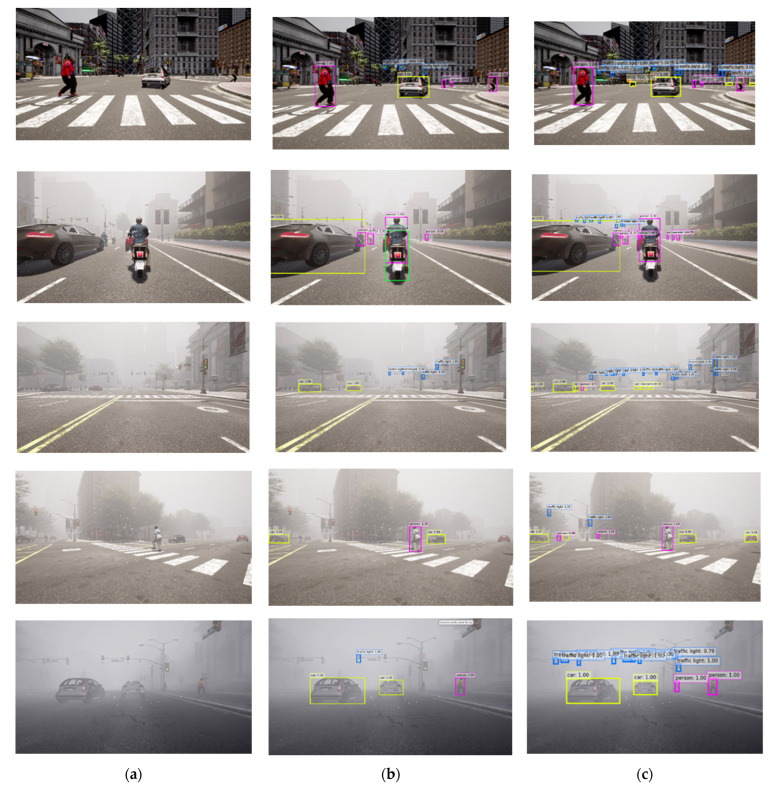
Comparison of the qualitative results of our CR-YOLOnet and baseline YOLOv5. (**a**) Input data with varying visibility and proximity: from clear conditions (**top**) to heavy fog (**bottom**). (**b**) Results of the medium YOLOv5 model trained on clear + fog. (**c**) Results of the small CR-YOLOnet model trained on clear + fog.

**Table 1 sensors-23-06255-t001:** Visibility distance values for clear, light, medium, and heavy fog conditions in the experiment.

Weather	Clear	Light Fog	Medium Fog	Heavy Fog
Experiment Value (m)	>1000	500–800	300–500	50–200

**Table 2 sensors-23-06255-t002:** Training parameters for the clear only, fog only, and clear + fog training sets.

Model	Model Size	Optimizer	Learning	Weight Decay	Batch Size	Momentum	Epoch
			Rate				
YOLOv5	Small	Adam	0.0001	0.00025	64	0.821	300
YOLOv5	Medium	Adam	0.0001	0.00025	64	0.821	300
YOLOv5	Large	Adam	0.0001	0.00025	64	0.821	300
CR-YOLOnet	Small	Adam	0.0001	0.00025	64	0.821	300
CR-YOLOnet	Medium	Adam	0.0001	0.00025	64	0.821	300
CR-YOLOnet	Large	Adam	0.0001	0.00025	64	0.821	300

**Table 3 sensors-23-06255-t003:** Performance comparison of our CR-YOLOnet and YOLOv5.

Model	Model Size	F1	Recall	Precision	mAP (0.5)	mAP Contrast	FPS
YOLOv5	Small	0.714	0.719	0.705	0.685	baseline	98
YOLOv5	Medium	0.692	0.776	0.738	0.771	↑0.086	46
YOLOv5	Large	0.756	0.813	0.792	0.795	↑0.110	25
CR-YOLOnet	Small	0.821	0.805	0.839	0.849	↑0.164	69
CR-YOLOnet	Medium	0.829	0.830	0.844	0.862	↑0.177	52
CR-YOLOnet	Large	0.861	0.885	0.914	0.896	↑0.211	27

**Table 4 sensors-23-06255-t004:** Comparison of detection *AP* for small, medium, and large object areas and *mAP* (0.5) using the large model size.

Model (Size: Large)	Trained on	Clear			Light Fog			Medium Fog			Heavy Fog			mAp (0.50)	Frame Rate (fps )
		APs	APm	APl	APs	APm	APl	APs	APm	APl	APs	APm	APl		
CR-YOLOnet	clear	0.928	0.957	0.989	0.903	0.928	0.936	0.808	0.817	0.871	0.791	0.727	0.833	0.815	22
YOLOv5	clear	0.833	0.856	0.877	0.698	0.727	0.783	0.679	0.693	0.728	0.611	0.505	0.613	0.745	28
CR-YOLOnet	fog	0.845	0.864	0.885	0.816	0.863	0.872	0.802	0.815	0.833	0.709	0.735	0.792	0.766	19
YOLOv5	fog	0.625	0.721	0.741	0.594	0.650	0.716	0.711	0.725	0.743	0.682	0.667	0.676	0.717	25
CR-YOLOnet	clear + fog	0.921	0.965	0.972	0.912	0.923	0.949	0.833	0.884	0.920	0.851	0.877	0.893	0.892	23
YOLOv5	clear + fog	0.795	0.869	0.883	0.717	0.739	0.755	0.748	0.769	0.806	0.740	0.632	0.677	0.798	25

**Table 5 sensors-23-06255-t005:** Comparison of detection *AP* for small, medium, and large object areas and *mAP* (0.5) using the medium model size.

Model (Size: Medium)	Trained on	Clear			Light Fog			Medium Fog			Heavy Fog			mAp (0.50)	Frame Rate (fps )
		APs	APm	APl	APs	APm	APl	APs	APm	APl	APs	APm	APl		
CR-YOLOnet	clear	0.912	0.938	0.975	0.847	0.864	0.914	0.758	0.768	0.822	0.740	0.676	0.784	0.770	36
YOLOv5	clear	0.820	0.830	0.858	0.698	0.689	0.737	0.643	0.636	0.665	0.557	0.451	0.560	0.632	65
CR-YOLOnet	fog	0.850	0.877	0.896	0.785	0.821	0.827	0.767	0.790	0.794	0.658	0.684	0.741	0.745	40
YOLOv5	fog	0.655	0.694	0.737	0.625	0.651	0.675	0.670	0.674	0.681	0.630	0.615	0.624	0.696	59
CR-YOLOnet	clear + fog	0.903	0.945	0.959	0.866	0.899	0.917	0.826	0.852	0.889	0.819	0.845	0.861	0.867	48
YOLOv5	clear + fog	0.791	0.802	0.843	0.718	0.703	0.746	0.717	0.711	0.744	0.689	0.579	0.625	0.765	54

**Table 6 sensors-23-06255-t006:** Comparison of detection *AP* for small, medium, and large object areas and *mAP* (0.5) using the small model size.

Model (Size: Small)	Trained on	Clear			Light Fog			Medium Fog			Heavy Fog			mAp (0.50)	Frame Rate (fps )
		APs	APm	APl	APs	APm	APl	APs	APm	APl	APs	APm	APl		
CR-YOLOnet	clear	0.841	0.877	0.911	0.732	0.751	0.855	0.693	0.701	0.743	0.627	0.679	0.714	0.751	68
YOLOv5	clear	0.785	0.798	0.819	0.661	0.684	0.730	0.587	0.598	0.628	0.433	0.528	0.530	0.572	92
CR-YOLOnet	fog	0.849	0.883	0.892	0.745	0.753	0.754	0.723	0.758	0.738	0.612	0.634	0.680	0.722	71
YOLOv5	fog	0.682	0.744	0.752	0.644	0.695	0.725	0.614	0.626	0.643	0.577	0.584	0.590	0.673	88
CR-YOLOnet	clear + fog	0.853	0.895	0.902	0.833	0.867	0.894	0.816	0.841	0.872	0.784	0.818	0.843	0.847	72
YOLOv5	clear + fog	0.695	0.765	0.792	0.674	0.745	0.751	0.645	0.661	0.692	0.546	0.585	0.638	0.682	98

**Table 7 sensors-23-06255-t007:** Comparison of detection *AP* per object class.

Model	Model Size	Clear			Light Fog			Medium Fog			Heavy Fog			mAR
		ARs	ARm	ARl	ARs	ARm	ARl	ARs	ARm	ARl	ARs	ARm	ARl	
CR-YOLOnet	Small	0.756	0.813	0.932	0.744	0.810	0.855	0.735	0.713	0.771	0.685	0.710	0.714	0.768
YOLOv5	Small	0.706	0.714	0.798	0.631	0.675	0.762	0.628	0.670	0.756	0.504	0.586	0.645	0.649
CR-YOLOnet	Medium	0.817	0.845	0.948	0.779	0.832	0.888	0.705	0.758	0.827	0.664	0.697	0.779	0.793
YOLOv5	Medium	0.694	0.719	0.818	0.673	0.701	0.803	0.672	0.678	0.759	0.554	0.665	0.709	0.710
CR-YOLOnet	Large	0.844	0.895	0.958	0.855	0.841	0.912	0.792	0.817	0.850	0.679	0.738	0.787	0.813
YOLOv5	Large	0.776	0.778	0.850	0.696	0.728	0.782	0.682	0.687	0.763	0.658	0.678	0.732	0.755

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
