# Peer review of "Deep Camera–Radar Fusion with an Attention Framework for Autonomous Vehicle Vision in Foggy Weather Conditions"

_sensors, 2023, doi:10.3390/s23146255_

Round 1

Reviewer 1 Report

The authors present a Deep Camera-Radar data fusion method to increase the robustness of object detection in adverse weather conditions. Overall, the work is complete, presenting a methodology and convincing simulation results. 

I have some concerns to be solved prior to publication:

1) The sentence "When it comes to object classification, radars fall short. Because radars can 45 only detect objects, they cannot classify what kind of object they are detecting since radar 46 detections are far too sparse [13,14]. The sparse nature of radar point clouds collected with 47 many vehicular radars (usually 64 points or less) might explain this [15]." is only partially true. A lot of research about imaging radar has been carried out in recent years, see [REF1,REF2,REF3,REF4], that led to cm-scale resolution coherent images. The Authors should discuss, or at least mention, imaging radars in the introduction with related references.

[REF1] M. Manzoni et al., "Motion Estimation and Compensation in Automotive MIMO SAR," in IEEE Transactions on Intelligent Transportation Systems, vol. 24, no. 2, pp. 1756-1772, Feb. 2023, doi: 10.1109/TITS.2022.3219542.

[REF2] Tebaldini, S.; Manzoni, M.; Tagliaferri, D.; Rizzi, M.; Monti-Guarnieri, A.V.; Prati, C.M.; Spagnolini, U.; Nicoli, M.; Russo, I.; Mazzucco, C. Sensing the Urban Environment by Automotive SAR Imaging: Potentials and Challenges. Remote Sens. 202214, 3602. https://doi.org/10.3390/rs14153602

[REF3] H. Wu and T. Zwick, “Automotive SAR for parking lot detection,” in Proc. German Microw. Conf., Mar. 2009, pp. 1–8.

[REF4] H. Iqbal, A. Löffler, M. N. Mejdoub, D. Zimmermann, and F. Gruson, “Imaging radar for automated driving functions,” Int. J. Microw. Wireless Technol., vol. 13, no. 7, pp. 682–690, Sep. 2021.

2) Where is the radar signal model? How the authors simulated radar data? Please improve by presenting the selected models

Author Response

1). We have discussed imaging radars in the introduction using the related references suggested by the reviewer.  See number line: 49

2). We have incorporated the radar detection model in section 3.2. See number line: 333

Reviewer 2 Report

In order to solve the problem of the AVs' s sensor performance degradation in fog, an improved YOLOv5-based multi-sensor fusion network is proposed in this paper. I find the work interesting and of certain guiding significance to the development of target detection. However, there are still some imperfections in the manuscript. I hope that my comments would be useful for improving the quality of the paper. Some of detailed comments are as follows.

1. In Section 3, Figure 5 has a cross-page phenomenon, which is recommended to be avoided.

2. The specific visibility of “clear, light, medium and heavy fog” in the paper is suggested to be described clearly, so that the specific performance of the proposed method can be better presented.

Minor editing of English language required

Author Response

1). We employed the Cross-Stage Partial (CSP) to increase our CNN's learning capabilities while decreasing the processing bottleneck and memory cost. It is also portable due to its low weight. We also made use of the SPP, which, regardless of the input size, produces a fixed-length output that is sufficient for our purposes. Unlike the sliding window pooling, which only uses one window size, SPP employs several window sizes, and uses multilayer spatial bins. The spatial model leverages the downsampling in convolutional layers giving max-pooling layers the attributes needed to build the spatial model.

2). We have included the specific visibility of “clear, light, medium and heavy fog” in Table 1. Please see line: 847.